# RNA Sequencing Analysis Reveals Divergent Adaptive Response to Hypo- and Hyper-Salinity in Greater Amberjack (*Seriola dumerili*) Juveniles

**DOI:** 10.3390/ani12030327

**Published:** 2022-01-29

**Authors:** Yuhao Peng, Hongjuan Shi, Yuqi Liu, Yang Huang, Renchi Zheng, Dongneng Jiang, Mouyan Jiang, Chunhua Zhu, Guangli Li

**Affiliations:** 1Guangdong Research Center on Reproductive Control and Breeding Technology of Indigenous Valuable Fish Species, Guangdong Provincial Engineering Laboratory for Mariculture Organism Breeding, Guangdong Provincial Key Laboratory of Pathogenic Biology and Epidemiology for Aquatic Economic Animals, Fisheries College, Guangdong Ocean University, Zhanjiang 524088, China; pengyuhaopyh@163.com (Y.P.); shihj@gdou.edu.cn (H.S.); lll12012021@163.com (Y.L.); zjouhy@126.com (Y.H.); zhengrenchi1@stu.gdou.edu.cn (R.Z.); jdn1987@163.com (D.J.); jiangmouyan@gdou.edu.cn (M.J.); zhu860025@163.com (C.Z.); 2Southern Marine Science and Engineering Guangdong Laboratory, Zhanjiang 524025, China

**Keywords:** transcriptome, carangidae, aquaculture, differential gene expression

## Abstract

**Simple Summary:**

The gill tanscriptomes of greater amberjack (*Seriola dumerili*) reared under different salinity stress were analyzed. The regulatory networks of salinity-related pathways were explored through Kyoto Encyclopedia of Gene and Genome (KEGG) pathway enrichment and bioinformatics analyses. This will be of great value in understanding the molecular basis of salinity adaptation in greater amberjack.

**Abstract:**

Salinity significantly affects physiological and metabolic activities, breeding, development, survival, and growth of marine fish. The greater amberjack (*Seriola dumerili*) is a fast-growing species that has immensely contributed to global aquaculture diversification. However, the tolerance, adaptation, and molecular responses of greater amberjack to salinity are unclear. This study reared greater amberjack juveniles under different salinity stresses (40, 30, 20, and 10 ppt) for 30 days to assess their tolerance, adaptation, and molecular responses to salinity. RNA sequencing analysis of gill tissue was used to identify genes and biological processes involved in greater amberjack response to salinity stress at 40, 30, and 20 ppt. Eighteen differentially expressed genes (DEGs) (nine upregulated and nine downregulated) were identified in the 40 vs. 30 ppt group. Moreover, 417 DEGs (205 up-regulated and 212 down-regulated) were identified in the 20 vs. 30 ppt group. qPCR and transcriptomic analysis indicated that salinity stress affected the expression of genes involved in steroid biosynthesis (*ebp*, *sqle*, *lss*, *dhcr7*, *dhcr24*, and *cyp51a1*), lipid metabolism (*msmo1*, *nsdhl*, *ogdh*, and *edar*), ion transporters (*slc25a48*, *slc37a4*, *slc44a4*, and *apq4*), and immune response (*wnt4* and *tlr5*). Furthermore, KEGG pathway enrichment analysis showed that the DEGs were enriched in steroid biosynthesis, lipids metabolism, cytokine–cytokine receptor interaction, tryptophan metabolism, and insulin signaling pathway. Therefore, this study provides insights into the molecular mechanisms of marine fish adaptation to salinity.

## 1. Introduction

Marine environmental factors, such as salinity, low O_2_ concentration, temperature, and pH value, influence the physiological and biological status of marine animals [1]. For instance, environmental stresses activate the sympathetic nervous system [2], the release of adrenaline and noradrenaline [3], and the hypothalamic–pituitary–interrenal axis in fish [4], thus causing the release of the steroid glucocorticoid hormones and other hormones for adaptation [5]. Although salinity enhances optimum fish growth, it can influence growth rate, immunity, antioxidant capacity, and lipid metabolism in fish [6,7]. For instance, salinity stress induces more active energy metabolism, including lipid metabolism and glycogen metabolism [8]. Salt stress environments trigger various metabolic changes in fish, thus enhancing fish adaptation to salinity [9,10]. Moreover, osmolality and water balance are energy-demanding processes maintained through osmoregulatory mechanisms, and they often alter survival, growth, and other physiological processes [11,12,13].

Osmotic stress responses associated with salinity changes activate molecular and physiological adaptations, such as variations in cell proliferation and differentiation of osmoregulatory organs [12], modulation of the expression and activity of ATPases [14], secondary activation of ion transporters [15], and structural proteins [16]. The osmoregulatory organs of fish include the gill, kidney, and digestive tracts. However, the gill plays the most crucial role because it has a large surface area and direct contact with the external environment [17,18]. Blood and water in gills are separated by just a few micrometers, thus facilitating the exchange of gases and allowing gill tissue to be exposed to environmental variation and pollutants [19]. Moreover, gills can balance the ion concentration in blood and acid base in freshwater and seawater-adapted fish [20,21,22]. Salinity changes can cause many lesions in gills, such as vascular congestion [8], lamellar fusion, mucosal cell and gill filament epithelium hyperplasia, loss of the structural integrity of pillar cells, and an increased number of chloride cells [23,24]. These lesions can change the crucial functions of gills and alter their morphological structure [19].

Some researchers have focused on the influence of salinity on the osmotic regulation of gills to investigate ionic and osmotic regulation in fish [9]. Furthermore, many studies have assessed the transcript expression profiles of teleosts, such as marbled eel (*Anguilla marmorata*) [25,26], spotted sea bass (*Lateolabrax maculatus*) [27], half-smooth tongue sole (*Cynoglossus semilaevis*) [28], Nile tilapia (*Oreochromis niloticus*) [15,29], silvery pomfret (*Pampus argenteus*) [30], Mozambique tilapia (*Oreochromis mossambicus*) [6], and Atlantic salmon (*Salmo salar*), after exposure to salinity changes [17]. As a result, some significantly differentially expressed genes and pathways related to salinity changes have been identified [31]. For instance, many classical ion transporters, including channels for amino acids [32], water [33], small solutes [34], calcium ions [35], sodium [36], chloride [37], and potassium [38], such as SLC (solute transport protein) families [39], AQP (aquaporin) families [40], NPY (neuropeptide Y receptor) families [41], and TRP (transient receptor potential) families are differentially expressed in the gills of teleosts under salinity changes [42]. Furthermore, researchers have also focused on pathways, including steroid biosynthesis, immune response, energy metabolism, apoptosis, cytokine–cytokine receptor interaction, and toll-like receptor signaling pathways [43].

The greater amberjack *Seriola dumerili* is a large, fast-growing species in the aquaculture industry worldwide with high commercial value [27,28]. Many researchers have assessed the factors limiting reproduction [44], ectoparasites [45], and weaning diets of greater amberjack [46]. However, no research has identified the optimum and limiting environmental factors for greater amberjack juveniles in captivity. This study explored the influence of different salinities on the gill transcriptome and gene expression of greater amberjack juveniles to identify and assess the genes with potential roles in salinity adaptation using RNA sequencing. Therefore, this study can provide a basis for understanding the physiology of greater amberjack and practical guidance for its commercial aquaculture production.

## 2. Materials and Methods

### 2.1. Ethics Statement

The experiments were conducted following the guidelines and regulations of the Animal Research and Ethics Committee of Guangdong Ocean University (NIH Pub. No. 85–23, revised 1996) and China’s laws and regulations on biological research. This study did not include any endangered or protected species.

### 2.2. Experimental Fish, Salinity Development, and Tissue Collection

A total of 80 greater amberjack juveniles (body length, 8.33 ± 0.45 cm and body weight, 6.38 ± 1.33 g) were used in this study. Before experiments, the greater amberjack juveniles were reared in tanks at 22 ± 1.0 °C in Donghai Island (Zhanjiang, Guangdong, China). They were randomly divided into four cylindrical 1000 L tanks (20 individuals per tank) at various salinities: 40, 30, 20, and 10 ppt (parts per thousand) groups. The salinity levels in the experiment were selected based on the previous salinity adaption study of greater amberjack larvals [47] and our unpublished data of juveniles. The fish in the control group were reared in natural seawater with a salinity of 30 ppt. The 40, 20, and 10 ppt groups were regulated using a commercial seawater salty crystal and aerated tap water. The fish were fed on commercial float bait twice a day at 9:00 and 19:00 for 30 days. Six fish were randomly selected from each group on 0, 15, and 30 days in our study, which was based on studies of other fishes, such as Asian seabass (*Lates calcarifer*, Bloch, 1790) [48], catfish (*Lophiosilurus alexandri*) [24], and cobia (*Rachycentron canadum*), under different salinity stress levels [31]. Unfortunately, the fish in the 10 ppt group were all dead within 10 days. The fish were then anesthetized using 100 mg/L tricaine methane sulfonate (MS 222; Sigma-Aldrich, St. Louis, MO, USA) and dissected. The gill tissues were immediately collected in centrifuge tubes containing 1 mL RNA stabilization reagent overnight, then stored at −80 °C for RNA extraction, sequencing, and gene expression analysis. The RNA of gill samples after 30 days were used for transcriptomic analysis, all gill samples after 15 and 30 days were used for qPCR verification.

### 2.3. Total RNA Extraction, Library Construction, and Illumina Sequencing

Trizol reagent (Invitrogen, Carlsbad, CA, USA) was used to extract total RNA from the gill tissues, following the manufacturer’s instructions. As previously described, the cleavage of gill tissue samples, RNA extraction, RNA purity, degradation, and contamination examinations were performed [49]. An Agilent 2100 bioanalyzer (Agilent Technologies, Palo Alto, CA, USA) was used to detect RNA integrity. Total RNA with an RNA integrity number (RIN) score >7 was used for sequencing.

TrueSeq RNA Sample Prep Kit (Illumina, San Diego, CA, USA) was used to obtain complementary DNA (cDNA) libraries from gill tissue, following the manufacturer’s instructions. Then, 3 µL of USER Enzyme (NEB, New England Biolabs, Palo Alto, CA, USA) was used with size-selected, adaptor-ligated cDNA at 37 °C for 15 min, followed by 5 min at 95 °C before PCR. PCR was performed using Phusion High-Fidelity DNA polymerase, universal PCR primers, and Index (X) Primer. The AMPure XP system was used to purify the PCR products. An Agilent Bioanalyzer 2100 system was used to assess the library quality. A cBot Cluster Generation System was used to cluster the index-coded samples via the TruSeq PE Cluster Kit v4-cBot-HS (Illumia, San Diego, CA, USA), following the manufacturer’s instructions. The library preparations were then sequenced on a HiSeq X-ten platform to generate the paired-end reads (150 bp).

### 2.4. Transcriptome Assembly and Functional Gene Annotation

The assembled *Seriola dumerili* genome (deposited in the DNA Data Bank of Japan (DDBJ) under accession numbers BDQW01000001-BDQW01034655 (Biosample ID: SAMD00083043_sdu_WGS.acclist.zip)) was used as a reference database for mapping reads [50]. The Illumina high-throughput sequencing platform was used to sequence the cDNA library, generating raw reads/data. The raw reads/data were filtered, then the adapter sequence and ploy-N (unable to determine base information) and low-quality (reads with <50% bases of quality value) reads were removed to obtain high-quality clean reads/data. Raw data and clean data were saved in FASTQ format. The Q20, Q30, GC-content, and sequence duplication levels of the clean data were measured. All the subsequent analyses were based on high-quality clean data.

BLASTx (version 2.2.26) (https://blast.ncbi.nlm.nih.gov/Blast.cgi, accessed on 29 December 2021) (E-value threshold, 1 × e^−5^) was used to annotate the gene function via the NR (https://www.ncbi.nlm.nih.gov/refseq/, accessed on 29 December 2021), Swiss-Prot (protein) (https://www.uniprot.org/;%20version%2020140317, accessed on 29 December 2021), KEGG (https://www.genome.jp/kegg/, accessed on 29 December 2021), COG (https://www.ncbi.nlm.nih.gov/research/cog/, accessed on 29 December 2021), and KOG (https://www.ncbi.nlm.nih.gov/research/cog-project/, accessed on 29 December 2021) databases, using the default setting. The clean library sequencing data were submitted to the National Center for Biotechnology Information (NCBI) Sequence Read Archive (SRA) (Bioproject number: PRJNA783747 (SRR17041711-SRR17041719)).

### 2.5. Analysis of Differential Expressed Genes (DEGs) Analysis

The greater amberjack gene expression levels were detected using the fragments per kilobase per million (FPKM) method. The DEGs in 20 vs. 30 ppt and 30 vs. 40 ppt groups were identified using the DESeq2 R package (version 1.16.1). DESeq2 was used to determine differential expression in digital gene expression data based on the negative binomial distribution model. Genes with a fold change ≥2 and a false discovery rate (FDR) <0.05 were considered DEGs. A KEGG pathway analysis was conducted, and DEGs with *p* < 0.05 were considered statistically significant [31]. In Donghai island, 30 ppt was the natural salinity of seawater. The DEGs in both 20 vs. 30 ppt and 30 vs. 40 ppt were identified and screened in order to clarify the effects of hypo-salinity and hyper-salinity seawater environments on gills of greater amberjack compared with the natural salinity.

### 2.6. Quantitative Real-Time PCR (QPCR) Validation

The expression patterns of DEGs in the RNA sequencing analysis were validated using qPCR. Sample collection, RNA extraction, and reverse transcription were performed as previously described [51]. RNA samples were obtained from the 20, 30, and 40 ppt groups. Each group had three replicates. The DEGs primers were designed using Primer5 software based on the assembled transcripts. A light CyclerTM96 (Roche, Indianapolis, IN, USA) was used for qPCR analysis, following the protocol of SYBR Green Real-Time PCR Master Mix (Takara, Tokyo, Japan). β-actin was used as a reference gene to normalize the expression levels [52,53,54]. The relative abundance of DEG mRNA transcripts was evaluated using the 2^−^^△△Ct^ method. The primer sequences for qPCR are shown in Appendix A.

### 2.7. Statistical Analysis

The relative mRNA expression levels and FPKM values are expressed as mean ± standard error (SE). The one-way ANOVA with Tukey’s post hoc test was used to evaluate the significant differences between 20 vs. 30 ppt and 30 vs. 40 ppt groups. The significance level was set at α = 0.05. The Statistical Package for the Social Sciences (SPSS) 16.0 (SPSS, Chicago, IL, USA) was used for all statistical analyses.

## 3. Results

### 3.1. Illumina Sequencing

The gill transcriptome can provide a valuable RNA resource for future analysis of greater amberjack adaptability to salinity and artificial culture. The HiSeq X Ten platform was used for RNA sequencing of gill samples at 20, 30, and 40 ppt. A total of 94.79, 93.66, and 78.83 million clean reads were obtained at 20 (G20), 30 (G30), and 40 ppt (G40), respectively, after quality control. The Q30 values and GC content of the clean reads were more than 95% and 49%, respectively. The high-quality reads were used for further analysis. The reads’ Q20 and Q30 values, GC content, and transcript numbers for each cDNA library are shown in Table 1.

### 3.2. Annotation and Functional Analysis of the Gill Transcriptome

In total, 23,972 genes were annotated. A total of 23,950 (99.91%); 22,838 (95.27%); 21,503 (89.70%); 16,505 (68.85%); 15,991 (66.71%); 14,525 (60.59%); 14,970 (62.45%); and 7382 (30.79%) were annotated in the NR, eggNOG, Pfam, Swiss-Prot, KOG, GO, KEGG, and COG databases, respectively (Table 2). The annotated genes provided the basis for further analysis of the specific molecular processes in greater amberjack.

The best BLAST results of reads were enriched for closely related fish species, including *Seriola lalandi* (6.24%), *Lates calcarifer* (2.09%), *Larimichthys crocea* (1.49%), *Stegastes partitus* (0.63%), *Oreochromis niloticus* (0.55%), and other species (4.98%) (Figure 1).

### 3.3. Identification and Analysis of Differentially Expressed Genes (DEGs)

A total of 417 (205 up-regulated and 212 down-regulated) and 18 (nine up-regulated and nine down-regulated) DEGs were identified in the G30 vs. G20 and G30 vs. G40 groups, respectively, using DESeq2 software, FDR-adjusted *p*-value < 0.05 and |Log2(fold change)| ≥ 1 (Figure 2). Heat maps of the clustered DEGs under hypo- and hyper-salinity stresses are shown in Figure 3. The top 20 DEGs in the G30 vs. G20 group and the top 18 DEGs in the G30 vs. G40 group are shown in Table 3.

### 3.4. DEG Annotation and Pathway Analysis

Some enriched KEGG pathways related to *Seriola dumerili* metabolism and molecular signaling pathway were identified, including steroid biosynthesis, cytokine–cytokine receptor interaction, porphyrin and chlorophyll metabolism, cytosolic DNA-sensing pathway, the intestinal immune network for IgA production, and tryptophan metabolism (Figure 4 and Figure 5). The top 10 enriched KEGG pathways of the DEGs under hypo- and hyper-salinity stresses are shown in Table 4.

### 3.5. Validation of RNA Sequencing Data by QPCR

Considering the top 20 most important salinity stress responses of DEGs identified in the transcriptomic analysis, which had been reported in the previous studies, nine genes were selected for RNA sequencing data validation using qPCR (Figure 6). The mRNA expression of *ebp*, *msmo1*, *nsdhl*, *sqle*, *lss*, and *ogdh* was significantly increased in the G20 group, while that of *ebp*, *msmo1*, and *sqle* was significantly decreased, compared with the G30 group. However, the mRNA expression of *nsdhl*, *lss,* and *ogdh* showed no difference between the G40 and G30 groups. The results were consistent with the RNA sequencing analysis results conducted using qPCR. Moreover, the mRNA expression of *edar*, *wnt4*, *slc25a48* were decreased in the G20 group. However, the mRNA expression of *edar* and *slc25a48* showed no difference between the G40 and G30 groups. Interestingly, the expression of these genes showed no differences at 15 days (Appendix A).

## 4. Discussion

### 4.1. Transcriptomic Analysis of Differentially Expressed Genes

Salinity alterations can lead to various physiological reactions to maintain homeostasis, including osmotic regulation, ion transports, and respiratory metabolism [55]. For instance, low (12 ppt) and high salinity water (32 ppt) can significantly induce some specific pathways in gills of silvery pomfret (*Pampus argenteus*), including calcium transport, neuroactive ligand–receptor interaction, NOD-like receptor signaling, Toll-like receptor signaling, and cytokine–cytokine receptor interaction pathway, indicating that salinity stress affects the immune system and osmotic pressure-regulated pathways [30]. Moreover, low salinity stress (6 ppt) affects ion transport, immune response, energy metabolism, and protein synthesis in marbled flounder (*Pseudopleuronectes yokohamae*) [9]. A total of 417 DEGs (205 up-regulated and 212 down-regulated) in the 30 ppt vs. 20 ppt group and 18 DEGs (nine up-regulated and nine down-regulated) in the 30 ppt vs. 40 ppt group were identified. KEGG analysis showed that the DEGs were mainly enriched in the cytokine–cytokine receptor interaction, apoptosis, steroid biosynthesis, and mTOR signaling pathways, similar to cobia (*Rachycentron canadum*) [31]. As discussed below, the DEGs were involved in several potential complex molecular biological processes in the gill.

### 4.2. DEGs Involved in Steroid Biosynthesis and Lipid Metabolism

Steroid hormones, such as epinephrine and cortisol, can influence the metabolic capacity of the gill [56]. Epinephrine induces glycogenolysis after exposure to stress, thus increasing the plasma glucose level, which provides energy for the target tissue, including gill, to transfer ions like Na^+^ [57]. In addition, cortisol can stimulate active Ca^2+^ uptake under asymmetrical conditions, thus regulating the tight junction morphology between pavement cells of euryhaline fish [56,58]. Herein, KEGG pathway analysis showed that some DEGs (*ebp*, *lss*, *sqle*, *nsdhl*, *msmo1*, *sc5d*, *dhcr7*, and *dhcr24*) were involved in the steroid biosynthesis pathways under the long-term hyper-salinity (40 ppt) and hypo-salinity (20 ppt) stresses. Moreover, salinity stress (from 20 to 40 ppt) decreased the mRNA expression levels of *sqle*, *msmo1*, and *ebp*. *Ebp* (Emopamil binding protein), also known as EBP cholestenol delta-isomerase, is essential in the sterol biosynthesis pathway [59]. Sterols are essential cell membrane components and transporters in many biofilms [60]. SQLE (squalene epoxidase) is rate limiting and the first oxygenation enzyme in cholesterol synthesis [61]. Lanosterol, especially cholesterol, is the upstream precursor of sterol biosynthesis in fungal steroids and animals [62]. Herein, hypo-salinity up-regulated *sqle*, *ebp*, and *lss* in the gills, indicating the stimulation of cholesterol synthesis in the gills. Studies have shown that *lss* and *dhcr24* are up-regulated in *O. niloticus* and *Rachycentron canadum* under hypo-salinity [31,63], consistent with this study.

Msmo1 is a key cholesterol biosynthetic enzyme. Nsdhl regulates adipogenesis via a synergized expression pattern with Msmo1 [64,65,66,67]. Herein, hypo-salinity up-regulated both *nsdhl* and *msmo1*, indicating that hypo-salinity can affect adipogenesis in greater amberjack. Moreover, hypo-salinity down-regulated *edar* and *tnfsf12* mRNA in greater amberjack. Studies have shown that EDAr (ectodysplasin A receptor), belonging to the tumor necrosis factor receptor (TNFr) superfamily, can regulate cell activities, such as differentiation, proliferation, maturation, and lipid metabolism, by binding to the ectodysplasin A1 (EDA1) [68,69,70,71]. A previous study showed that *tnfsf12* (TNF superfamily member 12) can inhibit lipid deposition in a dose-dependent manner without any cytotoxic effects. However, an agonistic antibody of the *tnfsf12* receptor can alleviate the repression [72]. Herein, hypo-salinity decreased the mRNA expression of *tnfsf12* in gill, indicating that lipid deposition is essential under low salinity stress. Previous reports have suggested that lipids are the energy source for euryhaline fish under osmotic stress [49,73]. Therefore, more lipids may be produced in greater amberjack under salinity changes by regulating the steroid biosynthesis related to lipid metabolism, adipogenesis, and lipid deposition, as revealed in other fish species [18]. The Ogdh enzyme is a key entry point for carbon into the Krebs cycle. Moreover, it affects all redox signals in mitochondria and cells [74,75]. Herein, hypo-salinity significantly increased the mRNA expression of *ogdh* mR, implying that adequate energy is needed in gills under salinity changes.

### 4.3. DEGs Involved in Ion Transport

Solute transport protein (solute carrier SLC) is the largest class of intracellular transport proteins, with over 300 members, mostly located in cell membranes. They mainly facilitate the transport of various substrates, including amino acids, nucleotides, glucose, and inorganic ions, across biological membranes [39]. Some of these proteins (*slc5a6a*, *slc4a1a*, *slc4a4a*) had different expression patterns between the hypo-salinity and control groups. The SLC4 gene family mediates HCO3^−^ extrusion and Cl^−^ uptake across cellular plasma membranes, thus regulating the cell volume and intracellular pH and stabilizing resting membrane potential through the regulation of cytoplasmic Cl^−^ [76]. Moreover, *slc4a1a* and *slc4a1b* are positively correlated with NKA (Na^+^/K^+^-ATPase) [77]. Herein, *slc4a1a* was down-regulated in the hypo-salinity group, while it was up-regulated in the hyper-salinity group, indicating the ion exchange in the gills.

Cystic fibrosis transmembrane conductance regulator (CFTR), an ABC transporter, acts as a channel across the cell membrane for transporting Cl^−^ into and out of cells. Hyper-salinity increases *cftr* expression in striped bass (*Morone saxatilis*) [78]. Moreover, *cftr* mRNA levels are significantly increased in gills of killifish (*Fundulus heteroclitus*) and Atlantic salmon (*Salmo salar*) under seawater conditions [79,80]. Similarly, this study showed that *cftr* was up-regulated in the hyper-salinity group, indicating the Cl^−^ transport changes.

Aquaporins (AQPs) are a family of integral membrane proteins that facilitate water transport across biological membranes along an osmotic gradient [81]. Thirteen AQP isoforms (AQP0-AQP12) have been identified in humans and rodents [40]. However, AQP4 has the most potential for high water permeability [82]. Herein, the expression of *aqp4* was significantly decreased in the hypo-salinity group compared with the control group, indicating the possible water permeability changes.

### 4.4. DEGs Involved in Immune Response

Previous studies have shown that the changes in fish immune status depend on the intensity and duration of the environmental stresses [83]. Herein, several DEGs (*wnt4*, *slc25a48*, *slc6a8*, *tlr5*, etc.) related to immune responses were found in the gill transcriptome of greater amberjack. Studies have reported that *wnt* genes enhance defense against pathogenic virus infection in the innate immune of *Litopenaeus vannamei* [84]. Moreover, Wnt4 protein can stimulate white blood cells and thymopoiesis in mice [85]. Herein, *wnt4* was significantly down-regulated in the hypo-salinity group, indicating that low salinity can affect the immune system of the greater amberjack.

Toll-like receptors (TLRs) are key pathogen pattern recognition receptors that control the host immune responses against pathogens by recognizing molecular patterns specific to microorganisms [86]. Previous studies have reported that *tlr5* plays a crucial role in the immune responses of turbot (*Scophthalmus maximus L.*) to the infections of various pathogens [87]. For instance, TLR5 stimulates the expression of proinflammatory, antibacterial, and stress-related genes by binding to bacterial flagellin, thus enhancing host defense against bacterial pathogens [88]. Moreover, flagellin increases *Tlr5* activity and the release of its downstream factor, IL−8 in mice [88,89]. Herein, the expression level of *tlr5* was decreased in the hypo-salinity group, indicating that low salinity inhibits *tlr5* activity on key adaptive functions, thus lowering efficient immune responses.

## 5. Conclusions

This work used transcriptome analysis to investigate the molecular changes in gills of greater amberjack (*Seriola dumerili*) under three different salinity concentrations (20, 30, and 40 ppt). The results provided large transcriptome data, abundant DEGs, and signaling pathways related to salinity adaptation. The signaling pathways analysis indicated that a complex molecular regulatory network is involved in metabolism, including steroid synthesis, lipid metabolism, tryptophan metabolism, ion transporters, and immune response for adaptation to salinity stress. Therefore, this study can provide insights into the molecular mechanisms of greater amberjack adaptation to salinity.

## Figures and Tables

**Figure 1 animals-12-00327-f001:**
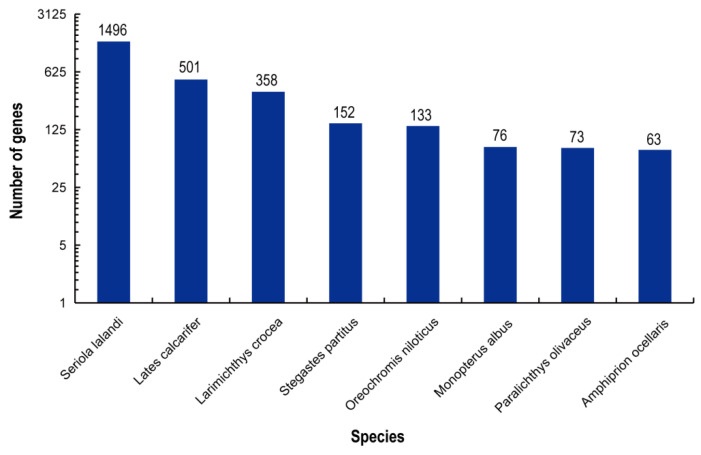
Species distribution in the Nr database. Vertical axis: the number of annotated sequences matching each species. Horizontal axis: the distribution of top species that match the annotated sequences.

**Figure 2 animals-12-00327-f002:**
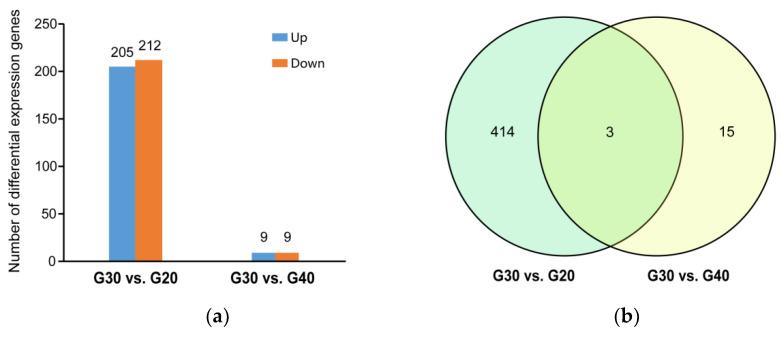
(**a**) The differentially expressed genes (DEGs) in the G30 vs. G20 group and the G30 vs. G40 group. The horizontal axis represents the G30 vs. G20 group and the G30 vs. G40 group, and the vertical axis indicates the gene numbers. The blue and orange colors represent the up- and down-regulated DEGs, respectively. (**b**) Number of DEGs and Venn diagram of the overlap of the groups.

**Figure 3 animals-12-00327-f003:**
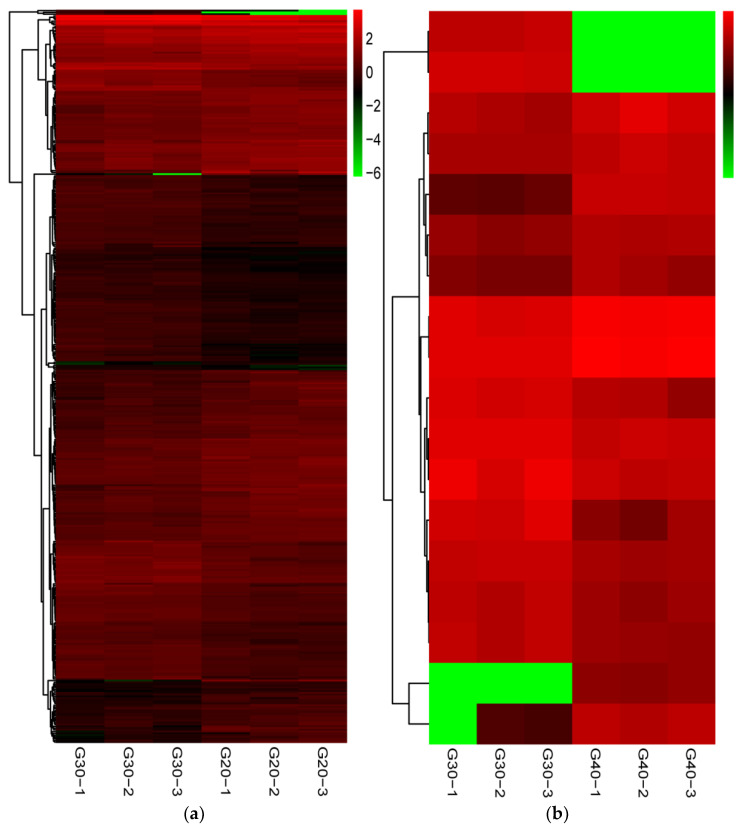
Expression heatmap of the DEGs in G30 vs. G20 group (**a**) and G30 vs. G40 group (**b**). Green and red squares indicate down- and up-regulation, respectively. The brighter colors indicate the significant fold changes.

**Figure 4 animals-12-00327-f004:**
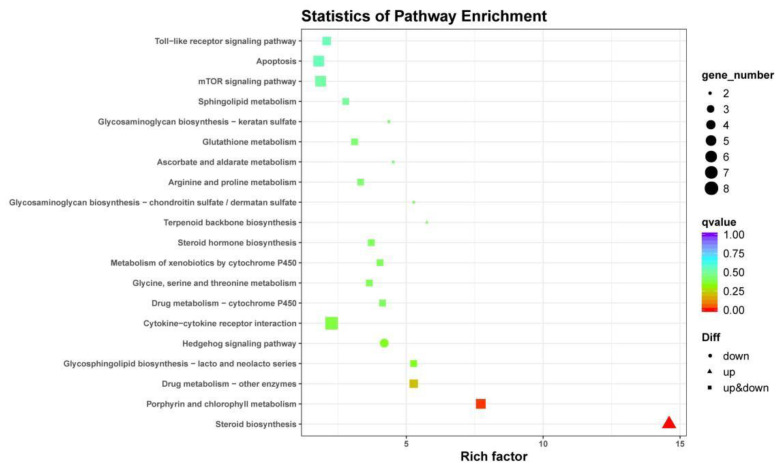
KEGG pathway enrichment analyses of DEGs in the G30 vs. G20 group. The size of each pathway represents the number of enriched targeted genes.

**Figure 5 animals-12-00327-f005:**
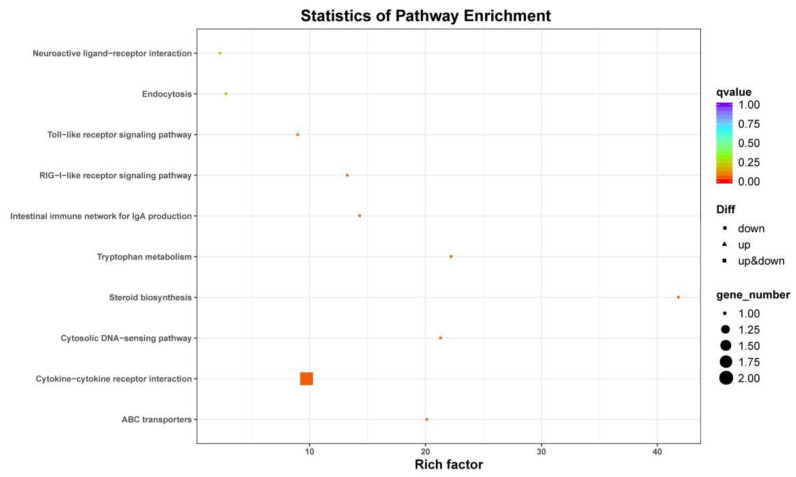
KEGG pathway enrichment analyses of DEGs in the G30 vs. G40 group. The size of each pathway represents the number of enriched targeted genes.

**Figure 6 animals-12-00327-f006:**
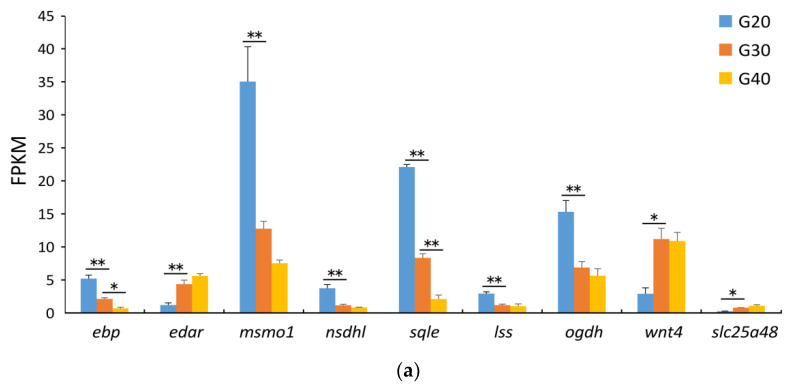
Expression of the nine validated genes in the G20, G30, and G40 group at 30 days. (**a**) The FPKM value detected by RNA sequencing. (**b**) The relative expression values measured using qPCR. The relative expression level of mRNA transcripts was detected using qPCR via the 2^−∆∆Ct^ method. Data are expressed as means ± SE (*n* = 3). The asterisks * and ** indicate statistical differences at *p* < 0.05 and *p* < 0.01, respectively, as determined by one-way ANOVA with Tukey’s post hoc test. β-actin was used as the reference gene.

**Table 1 animals-12-00327-t001:** Summary statistics of greater amberjack gill transcriptome sequencing data.

Samples ^a^	Clean Reads	Clean Bases	GC Contents (%)	Clean Reads Q20 (%) ^b^	Clean Reads Q30 (%) ^c^
G20-1	33,618,558	10,069,190,306	49.64	98.66	95.70
G20-2	30,542,362	9,149,565,768	49.59	98.63	95.63
G20-3	30,626,059	9,174,115,482	49.82	98.59	95.51
G30-1	29,118,900	8,723,118,946	49.98	98.65	95.68
G30-2	30,326,490	9,084,622,756	50.10	98.64	95.62
G30-3	34,209,823	10,247,870,004	49.86	98.61	95.59
G40-1	26,628,061	7,974,715,550	50.08	98.49	95.35
G40-2	29,931,391	8,964,260,212	49.86	98.63	95.72
G40-3	22,273,974	6,672,988,308	49.68	98.60	95.63

^a^ 1, 2, and 3: three independent biological replicates; ^b^ Q20: the percentage of bases with a Phred value > 20; ^c^ Q30: the percentage of bases with a Phred value > 30.

**Table 2 animals-12-00327-t002:** Greater amberjack gill transcriptome annotation statistics.

Category	Number	Percentage (%)
Total number of annotated genes	23,972	
Gene matches against GO	14,525	60.59
Gene matches against KEGG	14,970	62.45
Gene matches against KOG	15,991	66.71
Gene matches against Swiss-Prot	16,505	68.85
Gene matches against NR	23,950	99.91
Gene matches against Pfam	21,503	89.70
Gene matches against COG	7382	30.79
Gene matches against eggNOG	22,838	95.27

GO: gene ontology; KEGG: Kyoto Encyclopedia of Genes and Genomes; KOG: Eukaryotic orthologous group; NR: non-redundant; Pfam: Pfam protein families database; COG: clusters of orthologous groups; eggNOG: evolutionary genealogy of genes: non-supervised orthologous groups.

**Table 3 animals-12-00327-t003:** Top 20 DEGs in G30 vs. G20 group and top 18 DEGs in G30 vs. G40 group.

Gene Names	log2FC	Description
**G30 vs. G20**		
*Seriola_dumerili_newGene_5930*	2.80986	protein NLRC3-like
*Seriola_dumerili_newGene_10944*	2.51744	unnamed protein product
*LOC111226299*	2.19450	C-type lectin domain family 4 member F-like
*LOC111229131*	2.14477	growth/differentiation factor 8-like
*frmd3*	2.09078	FERM domain-containing protein 3
*LOC111231723*	1.99161	uncharacterized protein LOC111231723
*lyve1*	1.97051	lymphatic vessel endothelial hyaluronic acid receptor 1
*LOC111239963*	1.92871	solute carrier family 12 member 3-like
*slc5a6a*	1.81352	solute carrier family 5 member 6a
*Seriola_dumerili_newGene_13911*	1.75102	macrophage mannose receptor 1-like
*LOC111240141*	−3.71184	E3 ubiquitin-protein ligase TRIM21-like
*Seriola_dumerili_newGene_14148*	−3.58766	pol-like protein
*LOC111223147*	−2.40705	uncharacterized protein LOC111223147
*satb1a*	−2.38908	SATB homeobox 1a
*ribc2*	−2.22092	RIB43A domain with coiled-coils 2
*wnt7bb*	−2.09191	wingless-type MMTV integration site family, member 7Bb
*LOC111228626*	−2.01800	interferon-induced protein 44-like
*trabd2a*	−1.94610	TraB domain containing 2A
*Seriola_dumerili_newGene_12544*	−1.85641	Retrotransposable element Tf2 protein type 1
*LOC111220383*	−1.70808	gastrula zinc finger protein XlCGF57.1-like
**G30 vs. G40**		
*Seriola_dumerili_newGene_10944*	2.67769	unnamed protein product
*LOC111231293*	1.50771	C-X-C motif chemokine 10-like
*LOC111219635*	1.47708	intraflagellar transport protein 140 homolog
*slc4a4a*	1.24110	solute carrier family 4 member 4a
*cftr*	1.23478	cystic fibrosis transmembrane conductance regulator
*camk1a*	1.14853	calcium/calmodulin-dependent protein kinase type 1-like
*LOC111224523*	1.01042	kinesin-like protein KIF21A
*map7d2b*	1.00381	MAP7 domain containing 2b
*LOC111235326*	−3.05319	pleckstrin-like
*LOC111220915*	−2.00153	ladderlectin-like
*LOC111235291*	−1.29715	indoleamine 2,3-dioxygenase 2-like
*LOC111240189*	−1.15190	GTPase IMAP family member 7-like
*sqle*	−1.12522	squalene monooxygenase
*LOC111228808*	−1.08252	C-X-C chemokine receptor type 4-like
*Seriola_dumerili_newGene_4780*	−1.03479	nuclear receptor subfamily 1 group D member 1-like
*nr1d1*	−1.03175	nuclear receptor subfamily 1, group d, member 1
*LOC111236503*	−1.01894	granzyme A-like

**Table 4 animals-12-00327-t004:** Top 10 KEGG pathways of DEGs in G30 vs. G20 and G30 vs. G40 groups.

Pathway ID	Pathway Term	Gene Name
**G30 vs. G20**		
ko04060	Cytokine–cytokine receptor interaction	*LOC111221262*, *LOC111222833*, *LOC111224301*, *LOC111225535*, *LOC111230808*, *LOC111231248*, *edar*, *tnfsf12*
ko04210	Apoptosis	*LOC111218420*, *LOC111225343*, *LOC111228846*, *LOC111230905*, *bcl2l11*, *pik3cb*
ko00100	Steroid biosynthesis	*LOC111222971*, *ebp*, *lss*, *msmo1*, *nsdhl*, *sqle*
ko04150	mTOR signaling pathway	*LOC111220460, LOC111228846, LOC111229726, LOC111238290, pik3cb, wnt4*
ko00860	Porphyrin and chlorophyll metabolism	*newGene_6606, LOC111217420, LOC111219041, LOC111226879, LOC111237667*
ko04020	Calcium signaling pathway	*newGene_6757, LOC111224354, LOC111230822, nos1, ptafr*
ko04510	Focal adhesion	*LOC111228846, LOC111231768, itga9, pak5, pik3cb*
ko04340	Hedgehog signaling pathway	*LOC111228096, evc, hhip, ptch1*
ko00240	Pyrimidine metabolism	*LOC111217958, LOC111221687, cda, cmpk2*
ko04060	Cytokine–cytokine receptor interaction	*LOC111221262, LOC111222833, LOC111224301, LOC111225535, LOC111230808, LOC111231248, edar, tnfsf12*
**G30 vs. G40**		
ko00380	Tryptophan metabolism	*LOC111235291*
ko04144	Endocytosis	*LOC111228808*
ko02010	ABC transporters	*cftr*
ko04060	Cytokine–cytokine receptor interaction	*LOC111228808, LOC111231293*
ko04623	Cytosolic DNA-sensing pathway	*LOC111231293*
ko04672	Intestinal immune network for IgA production	*LOC111228808*
ko00100	Steroid biosynthesis	*sqle*
ko04620	Toll-like receptor signaling pathway	*LOC111231293*
ko04622	RIG-I-like receptor signaling pathway	*LOC111231293*

## Data Availability

The raw data of Illumina transcriptome have been submitted in the SRA under accession number PRJNA783747 (SRR17041711-SRR17041719).

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
