# Peer review of "RNA Sequencing Analysis Reveals Divergent Adaptive Response to Hypo- and Hyper-Salinity in Greater Amberjack (Seriola dumerili) Juveniles"

_animals, 2022, doi:10.3390/ani12030327_

Round 1
Reviewer 1 Report
Dear Authors
This is a very interesting paper about an important species for the aquaculture worldwide. I found the methods and the results quite good, and I have some comments about the content:
Line 23: Replace "breed" with "breeding"
Line 28: Avoid using abbreviations (RNA-Seq)
Line 30: "A total of 267,275,618 clean reads were generated." This is not so important to be added in the abstract
Line 45: I think "hypoxia" has to be replaced with "low O2 concentration".
Line 47: "The release"
Line 58: Remove "Significantly"
Line 73: Replace "aquatic biologists" with "researchers" maybe or a more appropriate word
Line 87: Remove "Some"
Line 90: Remove "belongs to the genus Seriola."
Line 95: "(40, 30, and 20 ppt)" this can be removed, it is also in the M&M section
Line 117: "Dead" instead of died
2.4: Please elaborate more on the bioinformatics analysis.
Line 159: Is it a fold change or logfold change?
Line 160 and throughout the manuscript: Avoid udsing "this study ...". Instead mention: A KEGG Pathway analysis was conducted etc
Section 2.6 and throughout the Manuscript: qPCR is enough to refer to this method
Why did you use only one reference gene? How did you check about its suitability?
2.7: "The one-way ANOVA was used to evaluate the significant 176 differences between 20 vs. 30 ppt and 30 vs. 40 ppt groups". Probably you did that with a post-hoc analysis?
Why did you compare only these pairs of groups and not all the possible pairs of groups? Please expain in the manuscript
Line 211: Replace "this study"
Lines 222-225: Apparently these are the best hits in the blast, not that these genes belong to these species. So please rephrase this section and make some changes in the corresponding figures
Line 283: Ion transport instead of ion transporter
Line 284: Induce instead of enrich
Line 338: Ion transport
Line 387: "For instance ... respectively" has to be removed, its not a conclusion
Author Response
Point 1: Line 23: Replace "breed" with "breeding"
Response 1: Thank you for your comment. This suggestion has been adopted; “breed” has been replaced by “breeding”. (Lines 23)
Point 2: Line 28: Avoid using abbreviations (RNA-Seq)
Response 2: Thank you for your comment. This suggestion has been adopted; “RNA-Seq” has been replaced by “RNA sequencing”. (throughout the Manuscript)
Point 3: Line 30: "A total of 267,275,618 clean reads were generated." This is not so important to be added in the abstract
Response 3: Thank you for your comment. This suggestion has been adopted, and the sentence has been deleted. (Lines 30)
Point 4: Line 45: I think "hypoxia" has to be replaced with "low O2 concentration".
Response 4: Thank you for your comment. This suggestion has been adopted, “hypoxia” has been replaced by “low O2 concentration”. (Lines 45)
Point 5: Line 47: "The release"
Response 5: Thank you for your comment. The sentence has been rewritten in the revised MS. (Lines 47)
Point 6: Line 58: Remove "Significantly"
Response 6: Thank you for your comment. The word has been deleted. (Lines 58)
Point 7: Line 73: Replace "aquatic biologists" with "researchers" maybe or a more appropriate word
Response 7: Thank you for your comment. This suggestion has been adopted; “aquatic biologists” has been replaced by “researchers”. (Lines 72)
Point 8: Line 87: Remove "Some"
Response 8: Thank you for your comment. The word has been removed. (Lines 86)
Point 9: Line 90: Remove "belongs to the genus Seriola."
Response 9: Thank you for your comment. The sentence has been rewritten in the revised MS. (Lines 89)
Point 10: Line 95: "(40, 30, and 20 ppt)" this can be removed, it is also in the M&M section
Response 10: Thank you for your comment. The sentence has been rewritten in the revised MS. (Lines 94)
Point 11: Line 117: "Dead" instead of died
Response 11: Thank you for your comment. The word has been rewritten in the revised MS. (Lines 119)
Point 12: 2.4: Please elaborate more on the bioinformatics analysis.
Response 12: Thank you for your comment. The section has been rewritten in the revised MS. (Lines 144-163)
Point 13: Line 159: Is it a fold change or logfold change?
Response 13: Thank you for your comment. It’s a fold change.
Point 14: Line 160 and throughout the manuscript: Avoid using "this study ...". Instead mention: A KEGG Pathway analysis was conducted etc
Response 14: Thank you for your comment. The sentence has been rewritten in the revised MS. (Lines 170)
Point 15: Section 2.6 and throughout the Manuscript: qPCR is enough to refer to this method
Response 15: Thank you for your comment. The word has been rewritten in the revised MS. (throughout the Manuscript)
Point 16: Why did you use only one reference gene? How did you check about its suitability?
Response 16: Thank you for your comment. The quantification of the β-actin gene was used as the reference gene, since its expression was already proved to be stable in greater amberjack Seriola dumerili at different sampling periods,or between wild and captive-reared fish (Pousis et al., 2017; 2019; Zupa et al., 2017). And in our experiment, the CT values of β-actin in each sample are very close.
References:
Zupa R, Rodríguez C, Mylonas CC, Rosenfeld H, Fakriadis I, Papadaki M; et al. Comparative study of reproductive development in wild and captive-reared greater amberjack Seriola dumerili (Risso, 1810). PLoS ONE 2017, 12(1), e0169645. doi:10.1371/journal.pone.0169645
Pousis, C.; Mylonas, CC.; De Virgilio, C.; Gadaleta, G.; Santamaria, N.; Passantino, L.; Zupa, R.; Papadaki, M.; Fakriadis, I.; Ferreri, R.; Corriero, A. The observed oogenesis impairment in greater amberjack Seriola dumerili(Risso, 1810) reared in captivity is not related to aninsufficient liver transcription or oocyte uptake of vitellogenin. Aquac Res 2017, 1-10. doi:10.1111/are.13453.
Pousis, C.; Rodríguez, C.; De Ruvo, P.; De Virgilio, C.; Pérez, JA.; Mylonas, CC.; Zupa, R.; Passantino, L.; Santamaria, N.; Valentini, L.; Corriero, A. Vitellogenin receptor and fatty acid profiles of individual lipid classes of oocytes from wild and captive-reared greater amberjack (Seriola dumerili) during the reproductive cycle. Theriogenology 2019, 140:73-83. doi: 10.1016/j.theriogenology.2019.08.014.
Point 17: 2.7: "The one-way ANOVA was used to evaluate the significant 176 differences between 20 vs. 30 ppt and 30 vs. 40 ppt groups". Probably you did that with a post-hoc analysis?
Response 17: Yes, you are right. We performed one-way ANOVA with Tukey’s post hoc analysis. The sentence has been rewritten in the revised MS. (Lines 189)
Point 18: Why did you compare only these pairs of groups and not all the possible pairs of groups? Please expain in the manuscript
Response 18: Thank you for your comment. The suggestion was adopted. We have added the explanation in the revised manuscript.
"In Donghai island, 30 ppt was the natural salinity of seawater. The DEGs in both 20 vs. 30 ppt and 30 vs. 40 ppt were identified and screened, in order to clarify the effects of hypo-salinity and hyper-salinity water environment on gills of greater amberjack, compared with the natural salinity." (Lines 171)
Point 19: Line 211: Replace "this study"
Response 19: Thank you for your comment. The sentence has been rewritten in the revised MS. (Lines 224)
Point 20: Lines 222-225: Apparently these are the best hits in the blast, not that these genes belong to these species. So please rephrase this section and make some changes in the corresponding figures
Response 20: Thank you for your comment. The section has been rewritten in the revised MS. (Lines 235-238)
Point 21: Line 283: Ion transport instead of ion transporter
Response 21: Thank you for your comment. The word has been rewritten in the revised MS. (Lines 299)
Point 22: Line 284: Induce instead of enrich
Response 22: Thank you for your comment. This suggestion has been adopted; “enrich” has been replaced by “induce”. (Lines 300)
Point 23: Line 338: Ion transport
Response 23: Thank you for your comment. The word has been rewritten in the revised MS. (Lines 354)
Point 24: Line 387: "For instance ... respectively" has to be removed, its not a conclusion
Response 24: Thank you for your comment. The sentence has been removed. (Lines 403)

Reviewer 2 Report
The author investigated the effect of different salt concentrations on gene expression in the gills of amberjack. However, it is not clear how much these salinities affect individuals physiologically because it is unpublished data (line 112). In fact, the authors use a 2-fold difference or greater and FDR of 0.05 or less to detect differences in gene expression levels, as described in Line 159, but these values are too low to take individual differences into account. In fact, most of the fold change values of the Top 20 DEGs are less than 8-fold at maximum, so we cannot deny the possibility that the salt concentration values used in this experiment do not have physiological significance. The author should explain more details why the salt concentrations were chosen in this experiment.
Line 115, Which samples on 0, 15, or 30 days were used in this transcriptome analysis and qPCR verification? Please describe why you chose those days and whether you checked the gene expression levels in the remaining samples. If you have data for the other samples, please show them in the revised manuscript.
In figure 6, what is the reasons to chose these genes? Not all of the genes are listed in Table 3.
In figure 6 and figS1 in qPCR results, ï½—hat value did the author use as 1 to indicate the relative value?
The sentence in line 268, the words of "expression" are redundant.
Author Response
Point 1: The author investigated the effect of different salt concentrations on gene expression in the gills of amberjack. However, it is not clear how much these salinities affect individuals physiologically because it is unpublished data (line 112). In fact, the authors use a 2-fold difference or greater and FDR of 0.05 or less to detect differences in gene expression levels, as described in Line 159, but these values are too low to take individual differences into account. In fact, most of the fold change values of the Top 20 DEGs are less than 8-fold at maximum, so we cannot deny the possibility that the salt concentration values used in this experiment do not have physiological significance. The author should explain more details why the salt concentrations were chosen in this experiment.
Response 1: Thank you for your comment. The salt concentrations (10, 20, 30 and 40 ppt) were chosen in this experiment, based on the survival rate and index greater amberjack larvals at different salinity levels from 10 to 40 ppt in the previous study (Chen et al.,1997).
"The salinity levels in experiment were selected based on the previous salinity adaption study of greater amberjack larvals [47] and our unpublished data of juveniles. " (Lines 112)
In our study, RNA sequencing and transcriptomic analysis showed the gene expression change are less than 8-fold at maximum, implied the a certain degree but not strong influences of salinity on greater amberjack juveniles physiology at 20 and 40 ppt, compared with 30 ppt. While 100% death rate was caused in 10 ppt within 10 days, indicating the sensitive of greater amberjack juveniles to the lower salinity environment.
References:
Chen, C.; Ji, R.; Huang, J.; He, H.; Liao, Z.; The relationship between the salinity and the embryonic, early larval development in Seriola dumerili. J. Shanghai Ocean University 1997, 6, 1
Point 2: Line 115, Which samples on 0, 15, or 30 days were used in this transcriptome analysis and qPCR verification? Please describe why you chose those days and whether you checked the gene expression levels in the remaining samples. If you have data for the other samples, please show them in the revised manuscript.
Response 2: Thank you for your comment. The suggestion was adopted. We have added the information of sampling in the revised manuscript.
"The RNA of gills samples at 30 days were used for transcriptomic analysis, all gills sample at 15, and 30 days were used for qPCR verification." (Lines 123)
We had searched the articles about salinity stress to commercial fish before our experiment, and the samples were often took at two weeks or one month. So we considered it could be the proper timing and decided to take samples on 0, 15, and 30 days. We have described it in the revised manuscript.
"Six fish were randomly selected from each group at 0, 15, and 30 days in our study, which was referred to other fishes, such as Asian seabass (Lates calcarifer, Bloch, 1790) [47], catfish (Lophiosilurus alexandri) [24] and cobia (Rachycentron canadum) [31], under different salinities stress ." (Lines 115)
The related reference are as bellow.
References:
Azodi, M., Bahabadi, M.N., Ghasemi, A. et al. Effects of salinity on gills’ chloride cells, stress indices, and gene expression of Asian seabass (Lates calcarifer, Bloch, 1790). Fish Physiol. Biochem. 2021, 47, 2027–2039. https://doi.org/10.1007/s10695-021-01024-6
Takata, R.; Mattioli, C.C.; Bazzoli, N.; Júnior, J.D.C.; Luz, R.K. The effects of salinity on growth, gill tissue and muscle cellularity in Lophiosilurus alexandri juvenile, a neotropical freshwater catfish. Aquac. Res. 2021, 52, 4064-4075.
Cao, D.; Li, J.; Huang, B.; Zhang, J.; Pan, C.; Huang, J.; Zhou, H.; Ma, Q.; Chen, G.; Wang, Z. RNA-seq analysis reveals divergent adaptive response to hyper- and hypo-salinity in cobia, Rachycentron canadum. Fish Physiol. Biochem. 2020, 46, 1713-1727.
In this study, we were mainly focused on studying the effects of different salinity on gills in greater amberjack juveniles . At present, we have not checked the genes expression levels in other tissues, it will carry out in the future.
Point 3: In figure 6, what is the reasons to chose these genes? Not all of the genes are listed in Table 3. In figure 6 and figS1 in qPCR results, what value did the author use as 1 to indicate the relative value?
Response 3: Thank you for your comment. The genes we chose in figure 6 were closely related to salinity adaptation of marine fish, combining with the top 20 DEGs identified by transcriptomic analysis with the DEGs reported in the previous studies. Thus, some of the identified important DEGs responsed to salinity stress, were not the top 20 DEGs. Above results indicated that the quality and validity of this transcriptome was good. We have described it in the revised manuscript.
"Consider of the top 20 and some important salinity stress responsed DEGs identified in transcriptomic analysis, which had been reported in the previous studies, nine genes were selected for RNA sequencing data validation using qPCR (Figure 6)." (Lines 275)
In our manuscript, the relative expression level of mRNA transcripts was detected using qPCR via the 2-∆∆Ct method. When we calculated it (2-(ΔCT-$ΔCTmax)=2-∆∆Ct), the maximum ΔCT value of each gene in gill sample was used as 1 (2-(ΔCTmax-$ΔCTmax)=20) to indicate the relative value.
Point 4: The sentence in line 268, the words of "expression" are redundant.
Response 4: Thank you for your comment. The word has been removed.

Round 2
Reviewer 1 Report
Thank you for addressing all of my comments and questions. Some minor changes in the language are needed.